# Machine learning-based prediction of diabetic retinopathy from pupillary abnormalities in a South Indian population

Janani Surya [1], S Tamilselvi[2], Maitreyee Roy[3], Sivaraj Chinnasamy[4], Rajiv Raman [5*], M Suchetha[2], Ganesh Rajendran[6]

1 University of New South Wales, Sydney, New South Wales, Australia, 2 Vellore Institute of Technology, Vellore, Tamil Nadu, India, 3 School of Optometry and Vision Science, UNSW Sydney, Sydney, New South Wales, Australia, 4 Corporeal Health Solutions Pvt Ltd, Chennai, Tamil Nadu, India, 5 Shri Bhagwan Mahavir Vitreoretinal Services, Medical Research Foundation, Sankara Nethralaya, Chennai, Tamil Nadu, India, 6 Vision Research Foundation, Sankara Nethralaya, Chennai, Tamil Nadu, India

* rajivpgraman@gmail.com

## Abstract

Diabetic retinopathy (DR) is a common complication of diabetes that can lead to vision loss. Early detection and prevention of DR is crucial to reduce the burden of this disease. The purpose of this study was to build a prediction model for DR using pupillary abnormalities as biomarkers. Pupillary parameters including Dark-adapted Baseline Pupillary Diameter (BPD), Amplitude of Pupillary Constriction (APC), Velocity of Pupillary Constriction (VPC), Amplitude of Pupil Re-dilatation after Maximum Constriction, and Velocity of Pupillary Dilatation (VPD) were collected and analyzed using machine learning algorithm including Support Vector Machine, Decision Trees, Artificial Neural Networks (ANN), Logistic Regressions, Random Forest, Naive Bayes Classifier. Utilizing ROC analysis and the Youden index, this study identified cut-off values for pupillary abnormalities to detect DR risk. The study found that ANN performed well with an accuracy of 0.807 (95% CI: 0.65–0.94) and AUC of 0.879 (95% CI: 0.71–0.98) in predicting DR using pupillary abnormalities as biomarkers. The findings of this research offer significant insights into the predictive value of pupillary abnormalities for DR, establishing a strong foundation for early intervention strategies. Particularly, the superior performance of ANN in detecting DR presents an opportunity to refine risk stratification and prevention approaches, potentially transforming the prognosis for individuals at elevated risk of this condition.

## 1. Introduction

Diabetic retinopathy is a significant microvascular complication of diabetes and a leading cause of blindness among working-age individuals [1]. Early detection and treatment of DR helps to prevent vision loss and improve patient outcomes. Pupillary abnormalities have been proposed as a potential biomarker for DR, with previous

**Data availability statement:** Due to ethical restrictions related to identifiable patient information, the dataset used in this study cannot be made publicly available. The Sankara Nethralaya Institutional Ethics Committee (Chennai, India) does not permit open public sharing of these data. De-identified data will be made available to qualified researchers upon reasonable request and after Ethics Committee approval and a Data Use Agreement. Data access requests may be sent to the institutional custodian: drpb@sankaranethralaya.org.

**Funding:** The author(s) received no specific funding for this work.

**Competing interests:** The authors have declared that no competing interests exist.

research demonstrating a relationship between changes in pupillary diameter and DR severity [2].Dynamic pupillometry is a rapid, non-invasive, and cost-effective method that has been widely explored in various disorders involving the Autonomic Nervous System (ANS), especially in diabetes [3–5].

Diabetes affects approximately 537 million people worldwide, with type 2 diabetes accounting for the vast majority of cases [2]. The current recommendation is an annual retinal screening of all these people with diabetes to detect vision threatening DR and treat them. However, systematic DR screening using retinal camera is extremely difficult to implement globally. In recent years, data mining techniques have been increasingly used to predict DR using various non-ocular parameters. Hosseini et al used ROC curves and logistic regression models to analyze the ability of a risk score to predict diabetic retinopathy in patients. The results showed that a risk score of 52.5 had an area under the ROC curve (AUC) of 0.704 and a sensitivity of 60% and specificity of 69% for predicting DR [6]. A mathematical model proposed by Aspelund et al., derived from epidemiological data, utilizes clinical factors such as diabetes type and duration, glycemic control indices, blood pressure, and current retinopathy status to predict the risk of sight-threatening retinopathy (STR). The performance of the model was evaluated using the AUC metric, which was 0.76 [7]. Likewise, Semeraro et al aimed to predict the risk of diabetic retinopathy using various statistical methods. The internal validation of the model resulted in a C-index value of 0.746 and a Gonen-Heller CPE of 0.683, indicating a good level of agreement between the observed and predicted occurrence of diabetic retinopathy [8].

We have previously demonstrated that pupillary dynamics are abnormal even in the early stages of diabetic retinopathy and progressively worsen with increasing disease severity [9]. Although numerous studies have reported abnormal pupillary dynamics in individuals with diabetes, suggesting autonomic dysfunction, the association between these alterations and the varying severity levels of diabetic retinopathy has not been thoroughly investigated. Kiziltoprak et al investigated the relationship between pupillary responses and the severity of diabetic retinopathy and found that patients with DR had lower values of amplitude and velocity of pupil contraction and dilation [10]. Cankurtaran et al evaluated the impact of diabetic retinopathy on automatic pupillometric measurements and found that non-proliferative diabetic retinopathy does not affect the results of both static and dynamic pupillary measurements [11]. However, proliferative diabetic retinopathy is associated with significant alterations in the results. Ortube et al measured pupillary constriction velocity and amplitude and found that these values were strongly correlated with the severity of diabetic retinopathy, not the duration of diabetes [12]. Karki et al suggest that while parasympathetic dysfunction in diabetic patients may be detected even without visible signs of diabetic retinopathy, sympathetic dysfunction may only be apparent when the retinopathy is moderate to severe [13]. Park et al proposed a method to identify pupillary light reflex (PLR) abnormalities in patients with non-proliferative diabetic retinopathy, particularly in the rods, cones, and melanopsin pathways. The experiments were conducted in both light-adapted and dark-adapted conditions. The results showed that steady-state pupil size decreased significantly in dark-adapted conditions for all

stages of diabetic retinopathy, and decreased only in patients with non-proliferative diabetic retinopathy in light-adapted conditions [14].

Compared to retinal imaging-based approaches, pupillometry offers a clinically practical and scalable alternative for DR screening, especially in settings with limited access to fundus photography. Its advantages include low cost, portability, rapid data acquisition, and minimal operator training, making it feasible for frontline or community-based deployment. Recent studies have also demonstrated that handheld or automated pupillometry devices can detect early neuro-optic dysfunction even before visible DR changes, underscoring their potential as adjunctive or triage tools [15–17].

There is a need to identify new methods to effectively detect the presence of vision-threatening DR. Herein, we aimed to derive and validate a novel non-invasive DR detection tool based on pupil dynamics for people with diabetes.

## 2. Methods

The present study is a secondary analysis of the dataset produced by our previously conducted non-interventional study "Pupillary Abnormalities with Varying Severity of Diabetic Retinopathy". The study design and research methodology has been described in detail elsewhere [9]. The study was approved by the institutional review board (Ethics Committee (Study code: 59–2007 P), Vision Research Foundation, and written informed consent was obtained from the subjects per the tenets of the Declaration of Helsinki. Data were originally collected in June 2007 and were accessed for research purposes from June 2007–2024 for the current analysis.

In patients attending a diabetic screening camp, 405 eyes of 244 subjects with diabetes mellitus over the age of 35 were examined. For the current study, they were divided into two groups; 147 diabetic eyes with no diabetic retinopathy (no-DR) as group 1 and the remaining 258 eyes with varying stages of diabetic retinopathy as group 2. Demographic data, detailed medical and ocular history, and a comprehensive eye examination were performed. A subset of 145 participants (102 no-DR and 43 DR) with complete data and clear labels was used for Machine Learning anslysis. Based on the severity of Diabetic retinopathy, one eye per participant was included to avoid inter-eye correlation.

After slit-lamp examination, subjects were dark adapted for 15 minutes for pupillography measurement. Pupil measurements were conducted using an infrared camera and a flashlight. Retinal photographs were captured after pupillary dilation with a Carl Zeiss fundus camera (Visucamlite, Jena, Germany). Each patient underwent 45°, four-field stereoscopic digital imaging covering the posterior pole, nasal, superior, and inferior regions. All retinal images were evaluated independently by two ophthalmologists in a blinded manner. The clinical grading of diabetic retinopathy was performed according to Klein's classification, based on the Modified Early Treatment Diabetic Retinopathy Study (ETDRS) system. The following parameters were obtained from the study, 1. Age, 2. Gender 3. Times for 1 Hippus were measured by the time between two Troughs/Crest before flash (in a sec). 4. Dark-adapted Baseline pupillary diameter was measured before exposure to the flash of light. 5. The amplitude of pupillary constriction to the flash of the light (in mm) and 6. The time (t1) taken was measured from the graph. 7. The velocity of pupillary constriction was calculated as the speed of constriction (distance divided by the time) in mm/sec. 8. The amplitude of pupil re-dilation after maximum constriction (in mm) and 9. time (t2) taken (in sec.) 10. The velocity of pupillary dilatation was calculated as the speed of re-dilatation (distance divided by time) in mm/sec. The parameters were analyzed using machine learning including Logistic Regression (LR) [18], K-Nearest Neighbors (KNN) [19], Artificial Neural Network (ANN) [20], Decision Tree (DT) [21], Naive Bayes (NB) [22], and Support Vector Machine (SVM) [23]. In response to an imbalance between the no-DR (0) and DR (1) groups in our dataset, we experimented with the SMOTE [24] technique to enhance model training. The dataset contains total 145 samples with 102 in normal category and 43 in DR category we applied SMOTE technique to increase this sample size in DR category. For machine-learning model development, one eye per participant based on DR severity was selected to avoid inter-eye correlation and ensure statistical independence of observations, as recommended for ophthalmic datasets [25].

A statistical analysis was conducted using StataMP version 17.0. All the data were expressed as mean ± SD. Pupillary parameters were compared between No DR and DR groups using the student T-test. To identifying optimal cut-off values

for these pupillary abnormalities in predicting DR, Receiver Operating Characteristic curves were utilized. The ROC analysis, guided by the Youden index, facilitated the determination of thresholds that maximize sensitivity and specificity. Following this, sensitivity, specificity, Diagnostic Odds Ratio (DOR), Positive Predictive Value (PPV), Negative Predictive Value (NPV), and accuracy for these cut-offs were meticulously calculated by comparing the prevalence of DR against these predictive markers. The statistical significance was considered at $P$ values < 0.05. For diabetic retinopathy prediction, we utilized machine learning algorithms implemented through the Keras library, alongside NumPy and Scikit-learn, in Python version 3.8. Keras, a high-level neural networks API, operates on top of TensorFlow, providing a streamlined and efficient framework for model development and testing. This setup allowed for the rapid execution of complex models while ensuring high accuracy in our predictive analysis.

## 3.  Model implementation and training

We have implemented various Machine learning models such as Random Forest, K Nearest neighbor, Logistic regression. Support vector machine, naïve bayes classifier, Decision tree and artificial neural network. Hyperparameter optimization was performed using a grid search strategy with five-fold stratified cross-validation to find the optimal combination of model parameters by minimizing overfitting as shown in Fig 1. In this process, the training data were partitioned into five folds, where four folds were used for training and one for validation in each iteration, to make sure balanced class representation across folds. The model achieving the highest average validation accuracy across folds was selected as the final classifier. To assess the robustness and reliability of the performance of the model, bootstrapping with 1000 resampling iterations was applied on the test set by repeatedly sampling the test set with replacement to estimate variability and 95% confidence intervals for accuracy and AUC metrics. In each iteration, a bootstrap sample is created by randomly drawing the same number of observations as the test set, but because sampling is done with replacement, some samples repeat while others are omitted by drawing many possible test sets from the same underlying population. A feedforward artificial neural network with two hidden layers was implemented using TensorFlow for binary classification. The first hidden layer consists of 256 neurons with ReLU activation, followed by the second hidden layer consisting of 128 neurons with the ReLU activation to add non-linearity and also to allow the network to learn complex patterns. The output layer is made up of one neuron with a sigmoid activation, which is suitable for binary classification as shown in the Fig 2. The model architecture and training parameters, including learning rate, number of hidden units, dropout rate, epochs, and batch size, were optimized through a grid search strategy combined with five-fold stratified cross-validation to ensure balanced and robust evaluation. The network was trained to run for 100 epochs with learning rate of 0.005 with batch size of 8, so that the model could converge optimally while also ensuring generalization. The justification for choosing these parameters has been presented in S2 Table.

| Parameter | Value |
|---|---|
| Learning Rate | 0.005 |
| Hidden Layers | 2 |
| Dropout Rate | 0 |
| Epochs | 100 |
| Batch Size | 8 |

## 4.  Results

Feature selection was performed using LightGBM [26] to identify the most discriminative pupillometry features for disease classification. The input features included physiological and temporal pupillary parameters such as age, sex, eye, time taken for 1 Hippus, mean crest, maximum trough, time taken for pupillary constriction to flash (seconds), amount of pupillary constriction to flash (millimeters), baseline pupillary diameter, time taken for pupillary dilation after flash (seconds), amount of pupillary dilation after flash (millimeters), and velocity of pupillary dilatation. All features were standardized

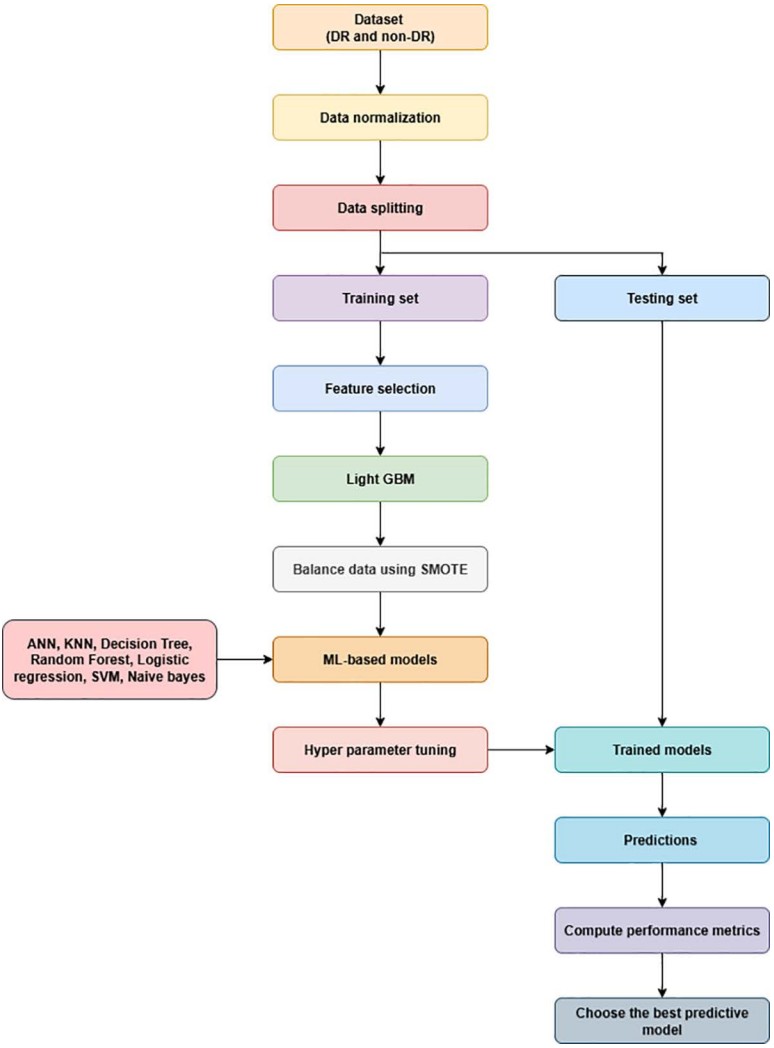

**Fig 1. Overall framework.**

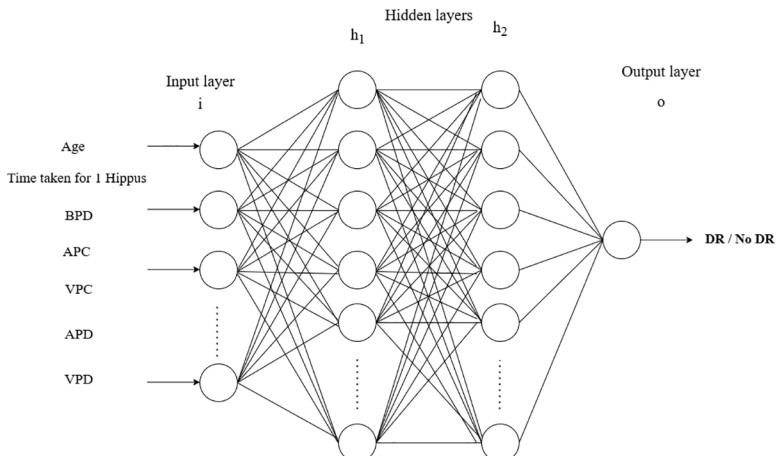

**Fig 2. Architecture of ANN.**

using z-score normalization for consistency, though LightGBM itself does not require feature scaling. A LightGBM classifier was then trained on the scaled features, and feature importance was extracted based on the gain metric, reflecting contribution of each feature in improving the classification model. The normalized importance scores were ranked to produce a final prioritized list of features. The resulting importance plot Fig 3 highlights top predictors including Amount of pupillary constriction to flash in mm, velocity of pupillary dilation (VPD), Time taken for 1 Hippus, time taken for pupillary dilation after flash, velocity of pupillary constriction (VPC), amount of pupillary dilation after flash, baseline pupillary diameter (BPD), age, and several other key features. These results underscore the significance of dynamic pupillary response metrics as influential biomarkers contributing to disease staging, with higher normalized gain values indicating stronger effects in the LightGBM classification model.

Table 1 presents the baseline pupillary dynamic parameters measured in all study participants for both eyes. The mean age was compared between the No DR group ($52.0 \pm 9.37$) and the DR group ($54.09 \pm 7.94$) found to be statistically significant ($P = 0.046$). There was a significant decrease in the pupillary dynamic parameters such as Time taken for 1 Hippus, BPD, VPC, amplitude of pupil re-dilatation after maximum constriction in the DR group in comparison with no DR group.

Table 2 displays the results of logistic regression and ROC curve analyses, detailing the diagnostic accuracy of various features in predicting Diabetic Retinopathy, based on full study participant measurements. For age, the estimated cut-off of 56 years has a Youden index of 0.177, correlating to a Diagnostic Odds Ratio (DOR) of 2.20, which suggests a significantly increased risk of DR. The threshold for the time taken for 1 Hippus is 0.65 seconds, showing a higher prediction accuracy with a DOR of 2.58. The cut-off for the Dark-adapted Baseline Pupillary Diameter is 4.50 mm, and for the Amplitude of Pupillary Constriction it is 4.57 mm, both yielding valuable predictive insights with DORs of 1.58 and 1.66, respectively. Notably, the Velocity of Pupillary Constriction cut-off is 2.14 mm/s, with DOR of 1.08. On the other hand, the

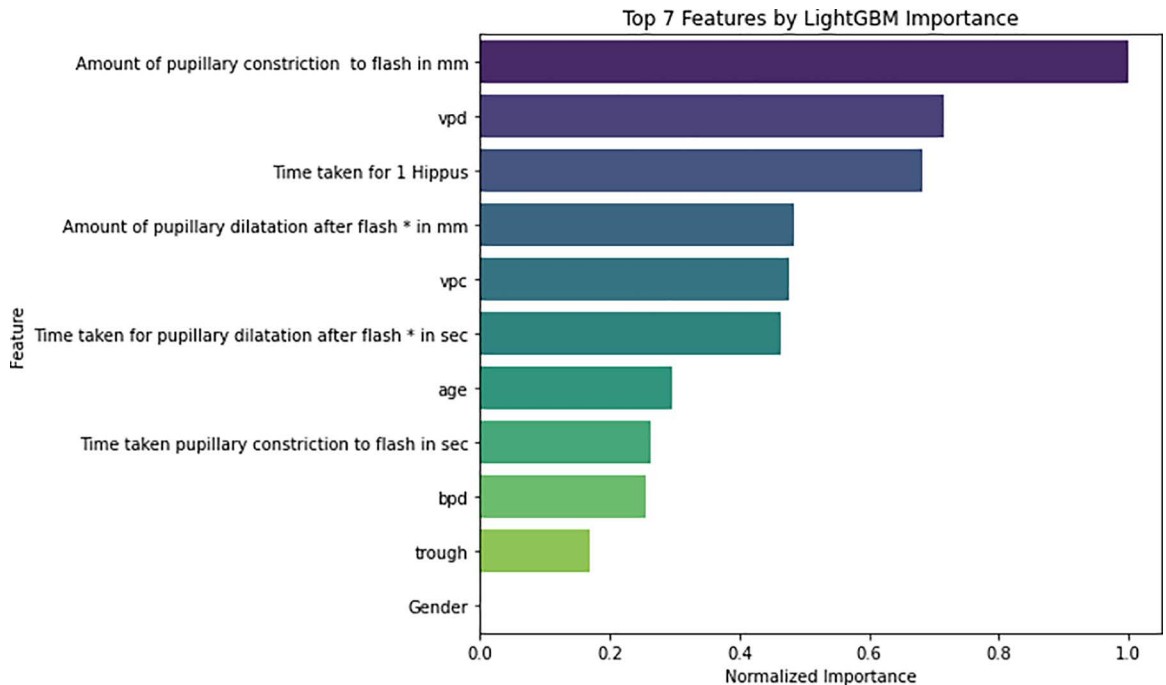

**Fig 3. Top pupillometry features identified by the Light Gradient Boosting Machine (LightGBM) importance scores for disease classification.**
The features include demographic factors such as age, gender and pupillary dynamics such as Hippus cycle time, dark-adapted baseline diameter, constriction and re-dilatation amplitudes, their respective times (t₁, t₂), and velocities of constriction and dilatation (mm/s).

**Table 1. Baseline pupillary dynamics and their association with diabetic retinopathy.**

| Features | NO DR | DR | P value |
|---|---|---|---|
| | N = 147 | N = 258 | |
| Age | 52.0 ± 9.37 | 54.09 ± 7.94 | **0.046** |
| Time is taken for 1 Hippus in secs | 0.74 ± 0.62 | 0.62 ± 0.25 | **0.000** |
| Dark-adapted Baseline pupillary diameter (BPD) in mm | 4.45 ± 0.63 | 4.31 ± 0.58 | **0.022** |
| The amplitude of pupillary constriction (APC) in mm | 4.46 ± 0.69 | 4.34 ± 0.91 | 0.174 |
| The velocity of pupillary constriction (VPC) (in mm/s) | 2.12 ± 0.57 | 1.94 ± 0.55 | **0.003** |
| The amplitude of pupil re-dilatation after maximum constriction (in mm) | 0.88 ± 0.36 | 0.82 ± 0.04 | **0.008** |
| The velocity of pupillary dilatation (VPD) (in mm/s) | 0.40 ± 0.16 | 0.42 ± 0.21 | 0.062 |

**Table 2. Cut-off values and diagnostic accuracy metrics for pupillary features in diabetic retinopathy prediction.**

| Features | Cutt off | Youden index | Diagnostic Odds Ratio (DOR) | Sen | Spec | PPV | NPV | Accuracy |
|---|---|---|---|---|---|---|---|---|
| Age | ≥ 56.00 | 0.177 | 2.20 | 0.44 | 0.74 | 0.41 | 0.76 | 0.65 |
| Time is taken for 1 Hippus in secs | ≥ 0.65 | 0.430 | 2.58 | 0.55 | 0.88 | 0.48 | 0.74 | 0.70 |
| Dark-adapted Baseline pupillary diameter (BPD) in mm | ≥ 4.50 | 0.113 | 1.58 | 0.53 | 0.58 | 0.35 | 0.75 | 0.57 |
| The amplitude of pupillary constriction (APC) in mm | ≥ 4.57 | 0.122 | 1.66 | 0.47 | 0.66 | 0.36 | 0.74 | 0.60 |
| The velocity of pupillary constriction (VPC) (in mm/s) | ≥ 2.14 | 0.018 | 1.08 | 0.67 | 0.34 | 0.30 | 0.71 | 0.44 |
| The amplitude of pupil re-dilatation after maximum constriction (in mm) | ≤0.69 | 0.471 | 2.64 | 0.54 | 0.93 | 0.50 | 0.73 | 0.70 |
| The velocity of pupillary dilatation (VPD) (in mm/s) | ≤0.76 | 0.351 | 3.10 | 0.42 | 0.93 | 0.53 | 0.73 | 0.71 |

Amplitude of Pupil Re-dilatation after Maximum Constriction, with a cut-off of 0.69 mm, shows a high accuracy rate in predicting DR, with a DOR of 2.64. The most accurate predictor is the Velocity of Pupillary Dilatation with a cut-off at or below 0.76 mm/s, demonstrating the DOR of 3.10, indicating its strong diagnostic power. These cut-off values, alongside their respective Youden index scores, define the most effective thresholds for each parameter to predict the presence of Diabetic Retinopathy, providing a measure of each feature's balance between sensitivity and specificity.

In Table 3, our analysis evaluates the performance of seven machine learning models using training, testing, and validation datasets. These models include Logistic Regression, K-Nearest Neighbors, Artificial Neural Network, Decision Tree, Naive Bayes, and Support Vector Machine. The performance is assessed by accuracy and Area Under the Curve, which represent predictive correctness and discriminative ability, respectively. The ANN model excels in the training set with an 90% accuracy and a robust AUC of 0.98. However, DT and RF models, while scoring perfectly in training, drop in performance in testing, highlighting a classic sign of overfitting.

**Table 3. Comparative performance metrics of Machine Learning Models across Training, Test, and Validation Sets.**

| Models | Training(N = 104) | | Test(N = 22) | | Validation(N = 19) | |
|---|---|---|---|---|---|---|
| | Accuracy | AUC | Accuracy | AUC | Accuracy | AUC |
| ANN | 0.99 | 0.99 | 0.80 | 0.88 | 0.87 | 0.97 |
| KNN | 1.0 | 1.0 | 0.67 | 0.79 | 0.80 | 0.83 |
| LR | 0.61 | 0.64 | 0.51 | 0.59 | 0.46 | 0.41 |
| SVM | 0.99 | 1.0 | 0.77 | 0.81 | 0.65 | 0.70 |
| NB | 0.61 | 0.62 | 0.64 | 0.62 | 0.38 | 0.33 |
| RF | 1.0 | 1.0 | 0.83 | 0.87 | 0.65 | 0.71 |
| DT | 0.98 | 0.99 | 0.58 | 0.59 | 0.57 | 0.59 |

The ANN's test accuracy of 80% and AUC of 0.88 are substantially higher than its counterparts, as shown in the Fig 4 the ROC curves illustrate, where ANN maintains a higher true positive rate across various decision thresholds. This indicates not only the model's accuracy but also its precision in classifying true cases of DR. The KNN model, despite its high validation accuracy of 80%, exhibits a lower test accuracy, calling attention to its limited ability to generalize beyond the training data.

When considering sensitivity and specificity, ANN consistently demonstrates reliable detection of true positives without a significant increase in false positives. Such balance is crucial in clinical applications where both false negatives and false positives carry significant consequences.

Table 4 presents a comparative performance analysis of several machine learning models for diabetic retinopathy prediction, reporting accuracy, AUC, precision, recall, and F1 score. The Random Forest model achieves the highest test accuracy at 0.83 (0.67–0.96), outperforming the Artificial Neural Network model, which performs with 0.8(0.64–0.93) test accuracy. Even though RF has high test performance, its validation accuracy is only 0.65, indicating overfitting to the training distribution. ANN demonstrates consistently strong performance with an AUC of 0.88, precision of 0.71, recall of 1.0, and F1 score of 0.83, outperforming most other models. Specifically, RF obtains the highest precision at 0.85, while ANN gives perfect recall score at 1.0. Compared to Logistic Regression, which achieves just 0.51(0.35–0.67) test accuracy, ANN outperforms significantly across all evaluated metrics. Overall, these results highlight that while RF performs better with high test accuracy, its lower validation performance indicates the importance of comprehensive model evaluation. Meanwhile, the ANN model demonstrates robust and reliable generalization for DR prediction. The results with SMOTE are shown in Table 3, while the results for the original imbalanced dataset are presented in S1 Table.

The prediction performance was evaluated using a three-way data split approach, which involves dividing the data into three parts: training, validation, and testing. The training data was used to train the model and learn the features, the validation data was used to adjust the model parameters, and the testing data was used to evaluate the final performance of the model. The model performance was evaluated in terms of sensitivity, specificity, and accuracy, and the results showed that the ANN model had a high level of performance, with a sensitivity of 0.73, a specificity of 0.83, and an accuracy of 0.80 (0.64–0.93). These results indicate the robustness and reliability of the ANN model for predicting DR using pupillary abnormal parameters.

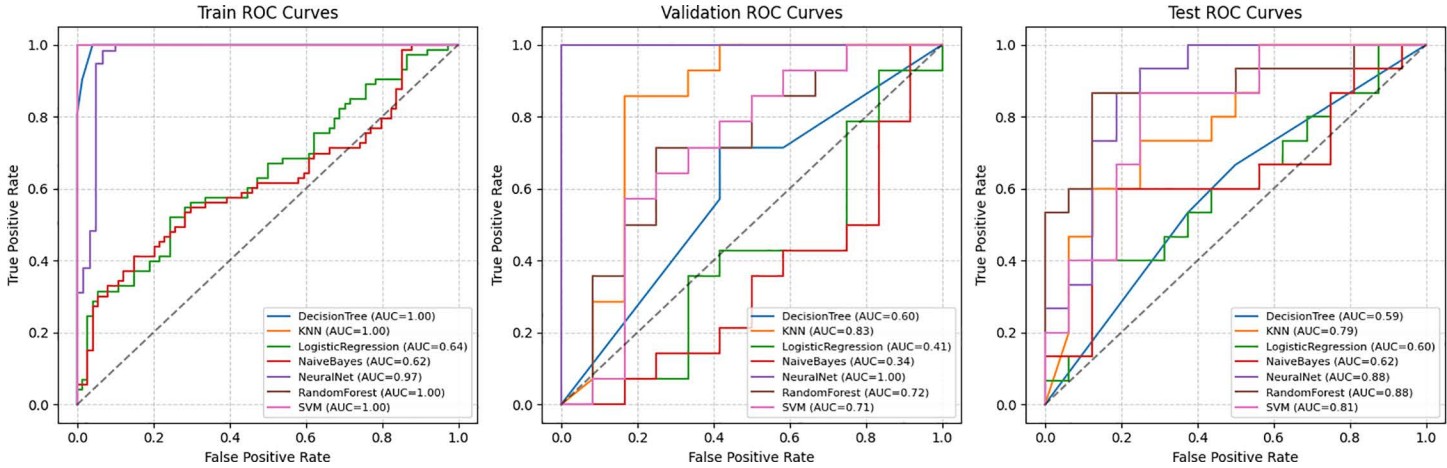

**Fig 4. Comparison of Receiver Operating Characteristic (ROC) curves for model performance across training, validation, and testing phases.** Models compared include Decision Tree Classifier, K-Nearest Neighbors (KNN), Logistic Regression, Naïve Bayes Classifier, Neural Network, Random Forest Classifier, and Support Vector Machine (SVM). The Area Under the Curve (AUC) values are reported for each model.

**Table 4. Test set performance metrics of machine learning models for DR prediction (95% CI reported for accuracy and AUC).**

| Models | Accuracy | AUC | Precision | recall | F1score |
|--------|----------|-----|-----------|--------|---------|
| ANN | 0.80 (0.64-0.93) | 0.88(0.71-0.98) | 0.71 | 1.0 | 0.83 |
| KNN | 0.67(0.51-0.83) | 0.79(0.61-0.92) | 0.61 | 0.86 | 0.72 |
| LR | 0.51(0.35-0.67) | 0.59(0.38-0.79) | 0.50 | 0.67 | 0.57 |
| SVM | 0.77(0.61-0.90) | 0.81(0.64-0.95) | 0.72 | 0.86 | 0.79 |
| NB | 0.64(0.48-0.80) | 0.62(0.42-0.82) | 0.7 | 0.47 | 0.56 |
| RF | 0.83 (0.67-0.960 | 0.87(0.72-0.98) | 0.85 | 0.8 | 0.83 |
| DT | 0.58(0.41-0.74) | 0.59(0.40-0.76) | 0.55 | 0.67 | 0.61 |

## 5. Discussion

Our study is unique in that it focuses on the prediction of Diabetic Retinopathy independent of retinal fundus imaging. To the best of our knowledge, this is the first investigation comparing seven distinct machine learning techniques for DR prediction using pupillometric data. Among these, the artificial neural network demonstrated the best performance, achieving a sensitivity of 0.73, specificity of 0.83, and overall accuracy of 0.80. Although these results are encouraging, they should be interpreted as preliminary given the exploratory nature and single-site design of the study. External validation and longitudinal follow-up are essential prior to clinical translation.

There has been extensive research and discussion in the literature regarding the relationship between pupillary abnormalities and diabetic retinopathy [6,9–14]. Previous studies have been developed to predict DR using data mining approaches with the clinical data such as age, gender, type of diabetes, duration of diabetes, HbA1c, hypertension and grade of diabetic retinopathy, medical history, blood tests and urine tests. Hosseini et al applied ROC curves for predicting DR. The study employed logistic regression models with diabetic retinopathy as the dependent variable, achieving an area under the ROC curve of 0.704, with 60% sensitivity and 69% specificity at a risk score threshold of ≥52.5 [6]. Tsao et al. developed predictive models for DR in patients with type 2 diabetes mellitus using various data mining techniques, including support vector machines, decision trees, artificial neural networks, and logistic regression. Their findings indicated that the SVM model outperformed the other algorithms, attaining an accuracy of 79.5% and an AUC of 0.839 under the percentage-split validation method [27]. Similarly, Hosseini et al. utilized logistic regression in combination with backward elimination as a feature selection strategy to predict the occurrence of DR. Evaluated on the training set of 3734 patients, they obtained AUC of 0.704, sensitivity of 0.603 and specificity as 0.694 [6]. Oh et al. incorporated sparse learning models to analyze health records for DR risk assessment in South Korea. By applying the least absolute shrinkage and selection operator (LASSO) [28] in combination with the Bayesian information criterion (BIC) for evaluating the internal validation cohort, the study achieved optimal performance metrics, with an AUC of 0.81, accuracy of 0.736, sensitivity of 0.774, and specificity of 0.727 [29]. Compared to these models, which rely on fundus-based or systemic features, our approach is distinctive in its use of infrared pupillometry integrated with a standard fundus camera. Fundus photography remains the clinical gold standard, and deep learning–based models using retinal images have demonstrated AUCs exceeding 0.90 in large datasets [30]. However, these systems require costly imaging infrastructure, trained personnel, and logistical support. In contrast, pupillometry offers advantages of portability, reduced cost, and rapid acquisition attributes that make it particularly suitable for deployment in low-resource or community screening contexts [15].

A key strength of our study lies in the moderate sample size and application of a diverse set of machine learning algorithms to analyze straightforward, non-invasive pupillary features. The use of an infrared pupillometer attached to a standard fundus camera enabled rapid, objective, and non-invasive measurement of pupil dynamics. Importantly, we used only one eye per participant in the machine learning models to avoid inter-eye dependency and maintain statistical

independence [25]. However, our device captured a limited set of features compared to high-end, research-grade pupil-lometers used in some prior studies.

While our model-derived cut-offs (e.g., BPD ≥ 4.50 mm) show statistical significance, their clinical translation requires further validation and contextual understanding. Prior studies have documented that pupillary constriction velocity and amplitude decline in diabetic patients, particularly with retinopathy, suggesting parasympathetic impairment correlates with DR progression [31]. Similarly, pupillary contraction metrics have been shown to correlate with retinal nerve fibre layer (RNFL) thickness, indicating that pupillometry may reflect retinal neurodegenerative changes [16]. Earlier work demonstrated that pupillary autonomic dysfunction can be detected before overt systemic symptoms, supporting the potential of pupillometry as an early biomarker [32]. Additionally, significant variation in pupillary response metrics across different DR severities has been reported, lending empirical support to the notion of stratified pupillometric thresholds [33].

Beyond these associations, it is important to recognize that pupillary responses can be influenced by factors such as age, ambient illumination, medication use, and concurrent ocular or systemic disease. Therefore, integration of pupillometric parameters with systemic risk markers like HbA1c, diabetes duration, and blood pressure and imaging-based metrics could enhance both specificity and predictive accuracy. In this context, standardized acquisition protocols and population-specific reference ranges will be essential for translation into clinical screening workflows or AI-driven triage systems.

It is also important to emphasize that pupillary abnormalities are not specific to diabetic retinopathy. Other ocular or systemic conditions, including diabetic autonomic neuropathy, glaucoma, optic neuropathies, macular degeneration, and medication effects, may similarly affect pupil dynamics. These potential confounders must be considered when interpreting pupillometric results in real-world settings.

Another limitation is the single-center, South Indian cohort, which restricts the generalizability of our findings to other ethnic, geographic, or healthcare populations. Factors such as iris pigmentation, ambient lighting, and cultural or environmental differences may influence pupillary responses. Larger, multi-centric, and multi-ethnic validation studies are warranted to assess reproducibility and robustness of these results.

Future studies could expand upon this work by incorporating a more diverse sample and explore whether integrating pupillary parameters with systemic variables such as HbA1c, diabetes duration, blood pressure or retinal imaging biomarkers can enhance predictive performance. If externally validated, pupillometry could serve as a rapid, non-invasive, and cost-effective adjunct or triage tool in DR screening programs, especially in low-resource or community-based settings where access to fundus photography or retinal specialists is limited.

This study presents preliminary evidence supporting the potential role of artificial intelligence models in predicting diabetic retinopathy based on pupillary abnormalities, without relying on retinal fundus images. Our results demonstrate that a simple, non-invasive pupillometric approach, analysed using machine learning, can detect patterns associated with DR, offering a promising adjunct for early risk identification. While the technique is cost-effective and relatively easy to deploy, particularly in resource-limited settings, these findings should be interpreted with caution, given the exploratory nature of the study. The clinical applicability of the identified cut-offs and their relevance across broader populations and settings remains to be validated.

By automating most analytic steps, the system offers rapid feedback that may aid in patient education and follow-up adherence. Moreover, our analysis suggests that examining relative sensitivity in pupillary response could contribute to the identification of neurological or ocular dysfunction in diabetes. Future research should focus on validating these findings in larger and more diverse cohorts, and on determining how pupillometry could complement traditional risk factors and imaging-based screening tools.

## Supporting information

**S1 Table. Comparative performance metrics of Machine Learning Models across Training, Test, and Validation Sets without using SMOTE.**
(DOCX)

**S2 Table. Effect of hyperparameter tuning on validation accuracy of the ANN model.**
(DOCX)

## Author contributions

**Conceptualization:** Rajiv Raman.

**Data curation:** Janani Surya, Rajiv Raman.

**Formal analysis:** Rajiv Raman, Ganesh Rajendran.

**Investigation:** Janani Surya, Sivaraj chinnasamy.

**Methodology:** Janani Surya, S Tamilselvi, Sivaraj chinnasamy.

**Project administration:** M Suchetha, Ganesh Rajendran.

**Software:** Ganesh Rajendran.

**Supervision:** Maitreyee Roy, Rajiv Raman, M Suchetha.

**Validation:** S Tamilselvi, Maitreyee Roy, Sivaraj chinnasamy, Rajiv Raman.

**Visualization:** Maitreyee Roy, Sivaraj chinnasamy.

**Writing – original draft:** Janani Surya, S Tamilselvi.

**Writing – review & editing:** Janani Surya, S Tamilselvi.

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
