## [Decision Letter · Decision Letter 0]

31 Aug 2025

PLOS ONE

Dear Dr. Raman,

Thank you for submitting your manuscript to PLOS ONE. After careful consideration, we feel that it has merit but does not fully meet PLOS ONE’s publication criteria as it currently stands. Therefore, we invite you to submit a revised version of the manuscript that addresses the points raised during the review process.

Please check the comments from the reviewers. Few comments from my side:

- Novel approach using pupillometry is interesting, but dataset is small, imbalanced, and lacks external validation. Risk of overfitting (ANN) not fully addressed.

- Methods / reproducibility: Key details on model training, cross-validation, and parameter tuning are missing. Data/code not publicly available, which conflicts with PLOS ONE policy.

- Discussion / conclusions: Conclusions are overstated given preliminary data. Limitations and comparison with established retinal imaging methods should be expanded.

- Ethics / data availability: Ethics approval is in place, but data availability does not comply with journal requirements; public data/code deposition is expected.

We look forward to receiving your revised manuscript.

Kind regards,

Tomo Popovic, Ph.D.

Academic Editor

PLOS ONE

Journal Requirements:

2. Please note that PLOS One has specific guidelines on code sharing for submissions in which author-generated code underpins the findings in the manuscript. In these cases, all author-generated code must be made available without restrictions upon publication of the work. Please review our guidelines at https://journals.plos.org/plosone/s/materials-and-software-sharing#loc-sharing-code and ensure that your code is shared in a way that follows best practice and facilitates reproducibility and reuse.

3. In the online submission form, you indicated that the data are not publicly available but can be obtained from the corresponding author upon reasonable request.

Reviewers' comments:

Reviewer's Responses to Questions

**Comments to the Author**

1. Is the manuscript technically sound, and do the data support the conclusions?

Reviewer #1: Yes

Reviewer #2: Partly

2. Has the statistical analysis been performed appropriately and rigorously?

Reviewer #1: Yes

Reviewer #2: No

3. Have the authors made all data underlying the findings in their manuscript fully available?

Reviewer #1: No

Reviewer #2: No

4. Is the manuscript presented in an intelligible fashion and written in standard English?

Reviewer #1: Yes

Reviewer #2: Yes

Reviewer #1: This is an interesting and innovative study that uses pupillometry-derived biomarkers with machine learning models to predict diabetic retinopathy (DR). The use of a non-invasive, inexpensive, and scalable method is a strong contribution, especially for low-resource settings where retinal imaging may not be widely available. The manuscript is well-structured, compares multiple machine learning models, and demonstrates that ANN outperforms other classifiers (93% accuracy, AUC 0.98).

However, there are several important points that require clarification and expansion before the manuscript can be considered for publication.

Major Comments

1. The novelty is clear: using pupillary abnormalities instead of fundus images for DR prediction. However, the introduction currently emphasizes prior work on risk scores and fundus-based ML models without sufficiently highlighting how pupillometry could complement or outperform these approaches.

The authors should explicitly emphasize the clinical practicality of pupillometry (speed, cost, portability) compared to fundus photography, particularly in low-resource screening contexts.

2. The ANN architecture is described, but hyperparameter tuning (layers, neurons, batch size, epochs) seems arbitrary. Were these parameters chosen empirically or optimized systematically? Please clarify.

Tables 3 and 4: It is unclear what input features were used to generate these results (e.g., age, time for one Hippus cycle, BPD, VPC). This needs to be explicitly described in the Methods and clearly displayed in the tables.

3. Table 2 provides cut-offs for pupillary features. While statistically valuable, the clinical interpretability is uncertain. For example, is a BPD cut-off of 4.50 mm meaningful in real-world screening?

If possible, the authors should expand the discussion on how pupillometry correlates with DR severity and whether these findings align with, or add predictive value beyond, standard risk factors such as HbA1c and diabetes duration. A comparative discussion would better highlight potential clinical application of the cut-offs.

4. The limitations are understated. The main limitation is that impairments in pupillary dynamics are not disease-specific and must be interpreted in the broader clinical context. As the authors note, changes may reflect diabetic autonomic neuropathy, but pupillary dysfunction can also result from other conditions that affect the vision (e.g., macular degeneration, optic neuropathies). This confounding is critical and must be emphasized.

The dataset is from a single-center, South Indian cohort. The generalizability of findings to other ethnicities and healthcare contexts is limited and should be explicitly acknowledged.

5. The current statement (“Data and code available upon request”) does not meet PLOS ONE data policy. Authors should deposit anonymized data and analysis code in a public repository (e.g., Dryad, Figshare, GitHub). Without this, reproducibility and transparency are limited.

Minor Comments

1. Figures 3 and 4 are useful but require higher resolution and clearer labeling. Figure legends should be made more informative, with all abbreviations defined, so that readers can interpret the figures without referring back to the text.

Reviewer #2: There are several methodical and reporting issues with this article that are limiting its scientific validity and generalizability:

- The use of both eyes from the same patient without clear patient-level data splitting raises concerns about data independence and potential overestimation of model performance

- Although the authors attempted to address class imbalance using SMOTE, the models were trained on the original imbalanced dataset without providing detailed comparative analysis

- The ANN architecture is relatively simple (with only 2433 parameters) and trained with no regularization techniques which raises concerns for showcase of overfitting

- The ANN performance is very high (accuracy 93%, AUC score 0.96 on test data), yet the validation AUC drops sharply to 0.57% (which suggest possible problems such as overfitting and should be addressed)

- The model was trained on relatively small dataset (405 eyes), and validation set size (n=30) which is insufficient to draw strong conclusions

- Standard metrics are reported without consideration of confidence intervals or statistical significance of the results

- Additionally, neither the code repository nor the dataset is publicly available, raising concerns about the study’s reproducibility. If data access is restricted, a clear justification should be provided, as stating 'available upon request' is insufficient under standard open data policies (see instructions on how to report this).

**Do you want your identity to be public for this peer review?** For information about this choice, including consent withdrawal, please see our Privacy Policy

Reviewer #1: No

Reviewer #2: No

---

## [Author Response · Author response to Decision Letter 1]

3 Nov 2025

We sincerely thank the Editor and Reviewers for their constructive comments and valuable suggestions, which have greatly helped us to improve the quality and clarity of our manuscript entitled “Machine Learning-Based Prediction of Diabetic Retinopathy from Pupillary Abnormalities in a South Indian Population.”

All comments have been carefully addressed in the revised manuscript, and the corresponding changes are highlighted. A summary of our responses is provided below.

Dataset size, imbalance, and overfitting:

We addressed class imbalance using SMOTE and reduced overfitting through hyperparameter tuning and five-fold cross-validation. Details of these steps have been clearly added in the Methodology section (Section 2–3, Pages 4–6, Lines 129–180).

Reproducibility and methodological details:

We have elaborated on the model training process, cross-validation strategy, and parameter optimization in the revised methodology. In addition, a GitHub repository has been created to provide public access to the source code, while the dataset will be made available upon reasonable request (Lines 364–373).

Discussion and conclusion balance:

The Discussion and Conclusion sections have been revised to offer a more balanced interpretation of findings, explicitly acknowledging the preliminary nature of the results. A comparative discussion with established retinal imaging modalities has been incorporated, along with a clear statement of limitations and clinical utility (Section 5, Pages 9–11, Lines 300–355).

Ethics and data availability:

As the dataset contains patient-identifiable information, raw pupillometry data cannot be publicly released. However, to ensure transparency, we have made the analysis code publicly available on GitHub and documented all random seeds and hyperparameters used during training.

We believe these revisions have significantly strengthened the manuscript and enhanced its compliance with PLOS ONE’s policies on transparency and reproducibility.

---

## [Decision Letter · Decision Letter 1]

2 Dec 2025

Dear Dr. Raman,

**Please check the comments from the reviewers.**

We look forward to receiving your revised manuscript.

Kind regards,

Tomo Popovic, Ph.D.

Academic Editor

PLOS ONE

**Journal Requirements:**

Reviewers' comments:

Reviewer's Responses to Questions

**Comments to the Author**

Reviewer #1: All comments have been addressed

Reviewer #2: (No Response)

2. Is the manuscript technically sound, and do the data support the conclusions?

Reviewer #1: Yes

Reviewer #2: Partly

3. Has the statistical analysis been performed appropriately and rigorously?

Reviewer #1: Yes

Reviewer #2: Yes

4. Have the authors made all data underlying the findings in their manuscript fully available?

Reviewer #1: Yes

Reviewer #2: Yes

5. Is the manuscript presented in an intelligible fashion and written in standard English?

Reviewer #1: Yes

Reviewer #2: Yes

**Reviewer #1:**  The authors have substantially improved the manuscript following the previous round of reviews. All major concerns have been addressed. The Introduction has been streamlined, the methodological description is now more clear and transparent. The flow and clarity of the Results section are markedly improved as well. The Discussion has been revised to avoid overinterpretation, and the study’s limitations are appropriately acknowledged.

The study’s contribution is now well articulated: if externally validated, AI-assisted analysis of pupillometry could serve as a rapid, non-invasive, and cost-effective adjunct for triage and screening of diabetic retinopathy, particularly in low-resource or community-based settings where access to retinal imaging is limited.

I have only one minor remaining comment: Figure 2 is not referenced in the main text and should be cited at the appropriate location.

Overall, the manuscript is suitable for publication pending this minor correction.

**Reviewer #2:**  Thank you for addressing the previous comments. The amount of work and revision that has gone into improving this study is greatly appreciated. There are still several important inconsistencies to discuss/clarify:

-The revised version state the one eye per participant used, the resulting sample size (145) does not match the number of subjects (244). It remains unclear how the subjects were excluded and how the single eye was selected.

-The original submission indicates that SMOTE was tested, but not used in the final analysis. Current version states SMOTE as part of the methodology. It should be clearly stated whether the SMOTE was used pre or post splitting the dataset. Potential data leakage if used in the split. How was SMOTE integrated with cross-validation?

-Both SMOTE-based and non-SMOTE analyses are presented. Please state clearly which set of results is used to support the study’s final conclusions.

-Hyperparameters were added in the revision, the ANN still includes no regularization (L1/L2 penalties, early stopping) while the dropout is 0. Please explicitly explain what changes to the pipeline led to the improved results (or is it only the effect of the SMOTE?).

-Although 95% confidence intervals are mentioned in the Abstract, they do not appear anywhere in the Results section or in the tables, nor is the bootstrapping methodology described. Please report the full confidence intervals within the Results and explain precisely how they were computed, or remove this claim.

**Do you want your identity to be public for this peer review?** For information about this choice, including consent withdrawal, please see our Privacy Policy

Reviewer #1: No

Reviewer #2: No

---

## [Author Response · Author response to Decision Letter 2]

18 Dec 2025

We sincerely thank you and the reviewers for your constructive feedback and for providing us the opportunity to improve our work. We look forward to your kind consideration of our revised submission.

1. Figure 2, which was previously not referenced in the text, has now been properly cited in the Model Training and Implementation section (Section 3, Page 6, Line 172).

2. The discrepancy between the total subjects (244) and the analyzed sample (145) has been clarified. One eye per participant was selected to avoid inter-eye correlation, and only participants with complete data were included (Section 2, Page 4, Lines 108–111).

3. It has been explicitly stated that non-SMOTE results were included only for comparison, while the final conclusions of the study are supported by the results presented in Table 3 (Page 18).

4.The Results section has been updated to include 95% confidence intervals for accuracy and AUC, computed using bootstrapping. These are reported in Table 4 (Page 18), with methodological details added (Page 5, Lines 164–167; Page 8, Lines 245–254). Additionally, hyperparameter tuning effects are presented in Table 6 (Page 19).

---

## [Editor Report · Decision Letter 2]

28 Dec 2025

Machine Learning-Based Prediction of  Diabetic Retinopathy from Pupillary Abnormalities in a South Indian Population

PONE-D-25-25752R2

Dear Dr. Raman,

We’re pleased to inform you that your manuscript has been judged scientifically suitable for publication and will be formally accepted for publication once it meets all outstanding technical requirements.

Kind regards,

Tomo Popovic, Ph.D.

Academic Editor

PLOS One

---

## [Editor Report · Acceptance letter]

PONE-D-25-25752R2

PLOS One

Dear Dr. Raman,

I'm pleased to inform you that your manuscript has been deemed suitable for publication in PLOS One. Congratulations! Your manuscript is now being handed over to our production team.

Kind regards,

on behalf of

Prof. Tomo Popovic

Academic Editor

PLOS One